

# Estimation of nitrogen leaching load from agricultural fields in the Puck Commune with an interactive calculator

Dawid Dybowski[1], Lidia Anita Dzierzbicka-Glowacka[1], Stefan Pietrzak[2], Dominika Juszkowska[2] and Tadeusz Puszkarczuk[3]

[1] Physical Oceanography Department, Ecohydrodynamics Laboratory, Institute of Oceanology Polish Academy of Sciences, Sopot, Poland
[2] Department of Water Quality, Institute of Technology and Life Sciences in Falenty, Raszyn, Poland
[3] Municipality of Puck, Puck, Poland

## ABSTRACT

**Background:** Nutrient leaching from agricultural fields is one of the main causes of pollution and eutrophication of the Baltic Sea. The quantity of nitrogen (N) leached from a particular field can be very different from the amount of N leached from other fields in a given region or even within a single farm. Therefore, it is necessary to estimate the quantity of N leached for each field separately.

**Methods:** An opinion poll has been conducted on 31 farms within the Puck Commune, which is approximately 3.6% of all farms located in this commune. Farmers provided data on the manner of fertilizing and cultivating crops on all their farms. For each field individually, on the basis of collected data, an estimated amount of the N leaching from the field has been determined.

**Results:** An interactive calculator to assist farmers in determining the quantity of N leaching from the agricultural field has been developed. The influence of factors shaping the amount of N leaching from a single field has been analyzed, and it has been determined that autumn plowing (specifically its absence) and the type of cultivated soil had the greatest average influence on this value in the studied sample.

**Discussion:** Due to the possible ways of reducing N leaching from agricultural fields, most of the studied fields were fertilized in an appropriate manner. However, in the studied sample there were fields for which the fertilization intensity significantly exceeded the recommended doses. In this context, a tool in the form of an interactive, easy-to-use N leaching calculator should help farmers to select appropriate doses and optimal fertilization practices.

Corresponding authors
Dawid Dybowski,
ddybowski@iopan.pl
Lidia Anita Dzierzbicka-Glowacka,
dzierzb@iopan.pl

## INTRODUCTION

The aim of agriculture, as well as any human economic activity, is to maximize efficiency. On the one hand, there is an attempt to maximize income (from the sale of plant to animal products). On the other hand, there is a attempt to reduce costs (fertilizers, equipment, activities). Modern large-scale agriculture cannot be imagined without fertilizers and pesticides. Each plant needs a certain amount of nutrients to grow. Increasing fertilizing

intensity may increase the potential yield. However, this yield reaches its maximum at some point and further increases in fertilizing intensity do not increase the yield but cause additional costs. Beside the obvious costs of fertilizer and all fertilizing-related activities of the farmer, there is an additional cost to the environment (*Álvarez et al., 2017*; *Heisler et al., 2008*; *Howarth, 2008*). Nutrient leaching from agricultural fields is one of the main causes of pollution and eutrophication of the Baltic Sea (*Elofsson, 2003*; *Ning et al., 2018*; *Voss et al., 2011*; *Savchuk, 2018*). In 2012, approximately 48,600 tonnes of nitrogen (N) (45.2% of total riverine N load from Poland) was delivered to the Baltic Sea as a result of farm activities in Poland (*Sonesten et al., 2018*). In view of the above, it is necessary to take in the measures of farms to reduce N leaching from agricultural soils. Among the possible measures used for this purpose, there should also be tools for quantitative control of nitrate losses due to leaching from agricultural fields. The choice of methods to counteract these losses depends on the recognition of their amount. In this context, it should be emphasized that the risk of N leaching is often considered by the N balance surplus. According to *Kupiec (2015)*, reference levels of N-surplus defining the risk of water hazards are quoted in various sources. Research results indicate that N-surplus can be a good predictor of groundwater nitrate pollution (*Wick, Heumesser & Schmid, 2012*; *Fraters et al., 2015*; *Huang, Ju & Yang, 2017*). However, the usefulness of this indicator for determining the risk of surface water nitrate pollution is not obvious when it is defined on the basis of data for the whole farm. Moreover, *Van Beek, Brouwer & Oenema (2003)* claim that estimates of N leaching to surface water based on data obtained for N balance "at the farm gate" level may be biased due to the heterogeneous distribution of N-surpluses on individual fields. Therefore, these authors postulate that N leaching to surface water from each agricultural field can be described as a function of soil surface balance surplus. *Lord, Anthony & Goodlass (2002)* examining the relationship between N balance "at the farm gate" and N leaching found that N-surplus was weakly or even negatively correlated with the concentrations (or loads) of nitrates in river waters. Thus, the use of N-surplus estimated by the "at the farm gate" method to determine the risk of surface water N pollution may not be appropriate.

Surely, the quantity of N leached from a particular agricultural field can be very different from the quantity of N leached from other fields in a given region or even within a single farm. Therefore, it is necessary to estimate the amount of N leaching for each field separately. The factors shaping the magnitude of N leaching are climate, soil type and management system. Each of these factors (except the climate) may vary for different fields within a given region. Main factors related to agriculture influencing the N leaching are:

- cultivation of inter-crops,
- the time of soil tillage,
- application of natural fertilizers, especially in autumn,
- annual doses of natural and mineral fertilizers.

The aim of the research presented in this article is to assess the approximate total N leaching from agricultural fields located in the Puck Commune. In the previous stage of

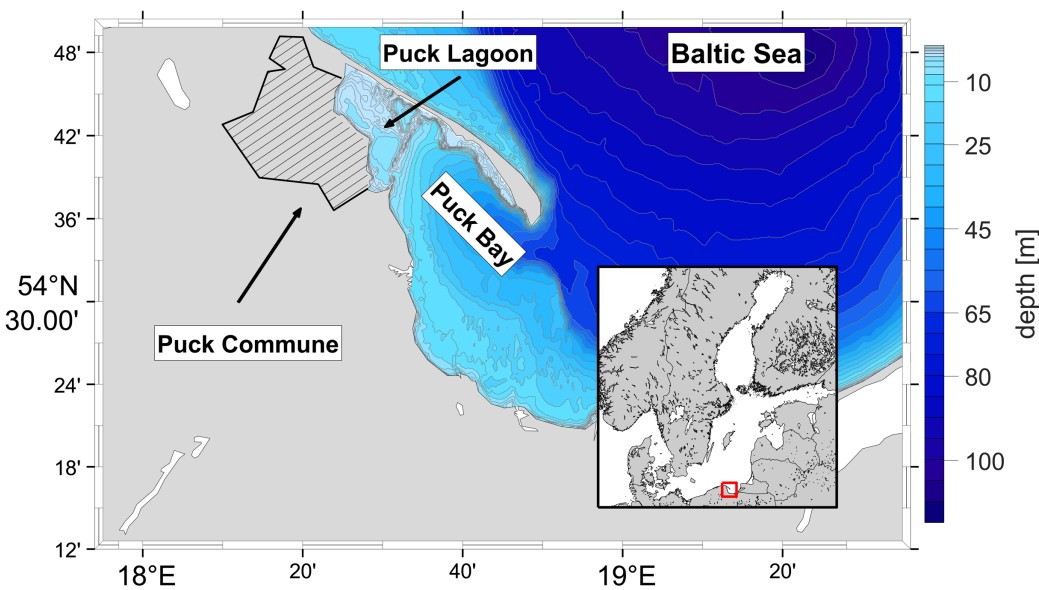

**Figure 1 Localization of the Puck Commune and the bathymetry of the Puck Bay as a part of Gdańsk Basin.**

work, an integrated agriculture calculator for establishing the balance of nutrients using the "At the farm gate" method was developed (*Dzierzbicka-Głowacka et al., 2019b*). The research was conducted as part of the project on modeling of the impact of the agricultural holdings and land-use structure on the quality of water in the Bay of Puck— Integrated information and forecasting Service "WaterPUCK" (*Dzierzbicka-Głowacka et al., 2019a*).

## METHODS

The Puck Commune is located in the north-eastern part of the Pomeranian Voivodeship (northern Poland), on the western shore of the Puck Bay which consists of the inner part called Puck Lagoon and the outer part of Puck Bay (see Fig. 1). The boundary between them runs from the Rybitwia Sandbank to the Cypel Rewski and has two straits within which there is an intensive water exchange between the Puck Lagoon and the outer part of the Puck Bay. Watercourses from Puck Municipality flow directly into the Puck Lagoon. Special attention should be paid to the quality of freshwater entering the Puck Lagoon. This water body is very sensitive to pollution due to geomorphological separation of the Puck Lagoon from the rest of the Puck Bay and its shallowness (the area of the Puck Lagoon is 30% of the entire Puck Bay and only about 6% of the water volume of the entire Puck Bay is located within Puck Lagoon). The ecohydrodynamic model of the Puck Bay called EcoPuckBay, whose hydrodynamic part has been validated (*Dybowski et al., 2019*), is in the final stage of preparation and is the high-resolution model describing the quality of the Puck Bay waters. In terms of climate, the area is located in a coastal region characterized by a high weather variability and, compared with other regions of Poland, colder summers and milder winters. The average temperature in summer is +13.5 °C and in winter +1.8 °C. The average annual precipitation does not exceed 700 mm.

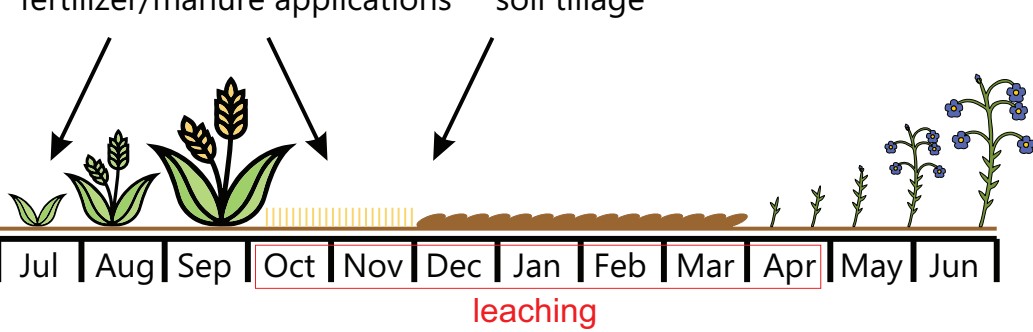

**Figure 2  N leaching period.**               

The prevailing winds are south and south-western. A characteristic phenomenon are breezes, as well as moving low-pressure areas causing strong winds, storms and heavy rainfalls. Snow cover lasts 40–60 days. The length of the growing season reaches 215 days (*Gawlikowska et al., 2009*). The multi-year annual average of solar surface irradiance is about 110 $Wm^{-2}$, while the multi-year summer average is two times higher (*Klimat w Polsce, 2014*).

The method for estimating the quantity of N leached from the agricultural field used in this article has been adapted to Polish conditions by *Aronsson & Ulén (2013)* from *Aronsson & Torstensson (2004)* and *Hoffmann (1999)* with support from a scientific team from the Institute of Technology and Life Sciences in Falenty.

It has been assumed that the growing season lasts from 1st September of the previous year to 31 August of the current year. N leaching begins at the beginning of autumn, immediately after harvest and continues throughout the winter until the start of the plant growing season (see Fig. 2). The amount of leaching is a result of all the activities undertaken in the previous crop season, and the main factors are:

- the type of crop grown in the summer before the start of the current season,
- methods of plant fertilization and soil tillage after harvesting.

## Factor A—soil type and the impact of the climate

In soils with high cation-exchange capacity, the nutrients supplied with fertilizers (e.g., ammonium, potassium, magnesium) are not leached into the soil profile and groundwater but are exchanged from the sorption complex during plant development. The sorption capacity is also of key importance for limiting the migration and bioavailability of trace metals. In soils with excessive metal contamination (e.g., cadmium or lead), a high sorption capacity reduces the leaching and transfer of metals to the food chain.

The total N content of the soil is most dependent on humus content, mineralization conditions shaped by water conditions of the soil and climate, the type of bedrock, the direction and degree of advancement of the soil-forming process. In soils used for agricultural purposes, an important factor shaping the N content is the level of organic and mineral fertilization and crop rotation, especially the share of legumes binding free N

**Table 1 Basic N leaching (kg N·ha$^{-1}$) with different amounts of precipitation and from different soil types.** Source: *Aronsson & Ulén (2013)* based on *Aronsson & Torstensson (2004)* and *Hoffmann (1999)*.

| Precipitation (mm) | Sandy soil | Loamy soil | Clay soil | Organic soil |
|---|---|---|---|---|
| 500–700 | 30 | 20 | 15 | 30 |
| 700–1,000 | 40 | 30 | 20 | 40 |

from the air (*Lityński & Jurkowska, 1982*). The vast majority of the N in the soil is incorporated into the organic part of the solid phase of the soil. N occurs in soil in the form of mineral and organic compounds and as molecular N in soil air. It comes either from fertilization or from microbiological processes—ammonification and nitrification. The average mineral nitrogen content in soils in Poland ranges from about 6 to 11 mg N kg$^{-1}$ depending on the soil type. The most easily available form of N for plants is nitrate nitrogen. It varies considerably during the year depending on the weather conditions, the intensity of uptake by the plants and the amount of fertilizer applied (*Fotyma, Kęsik & Pietruch, 2010*).

The majority of N transformations are determined by the activity of soil microflora. The transformations of N compounds in the soil have a significant influence on the overall natural N cycle. The balance of these transformations determines the conditions of N nutrition of plants and also determines the extent to which they use N fertilization. N mineralization consists of a set of processes leading to the formation of ammonia or ammonium N. This is essential for plants, as ammoniacal N is a form directly absorbed by their root system and is easily converted into nitrates, which are even more easily used by plants. N losses in the soil are caused by crop cultivation, water and wind erosion and denitrification processes. N in nitrate form can be denitrified or leached if it is not taken up by the plants. Because nitrate ions are highly mobile in soil, they move like water, that is, both upward (if evapotranspiration is higher than precipitation) and downward (otherwise). Therefore, a real threat of nitrate leaching occurs only during the winter half-year, because in the summer half-year, that is, when the temperature exceeds 5 °C, evapotranspiration dominates and water moves from deeper layers to the surface. Therefore, in the summer half-year nitrate leaching is recorded only after significant rain event. Nevertheless, with high nitrate content in the soil, there is a risk of eutrophication of surface water (especially the first layer) and therefore rational fertilizer management should be applied in accordance with the guidelines of the Code of Good Agricultural Practices (*Department for Environment, Food & Rural Affairs, 2009*) or the Nitrate Directive (*The Council of the European Communities, 1991*).

The method used in this article defines the concept of so-called basic leaching as the equivalent of N leaching losses in conventional cereal cultivation, under conditions of sustainable mineral fertilization and mid-autumn plowing, but without the use of organic fertilizers (*Aronsson & Ulén, 2013*). When determining the basic leaching value, the soil type and average precipitation in the region have been taken into account (Table 1).

Table 2 **Factors affecting basic leaching depending on the crop in the previous year.** Source: *Aronsson & Ulén (2013)* based on *Aronsson & Torstensson (2004)* and *Hoffmann (1999)*.

| Crop in the previous year | Factor |
|---|---|
| Cereal | 1.0 |
| Cereal followed by winter wheat | 0.9 |
| Cereal followed by winter oilseed | 0.8 |
| Cereal and oilseed with undersown catch crops | 0.7 |
| Cereal and oilseed with catch crops sown after | 0.9 |
| Cereal with undersown ley (grass and legumes) | 0.7 |
| Oilseed | 1.2 |
| Oilseed followed by winter wheat | 1.1 |
| Oilseed with undersown catch crops | 0.7 |
| Oilseed with catch crops sown after | 0.9 |
| Finalising ley without plowing | 0.6 |
| Ley plowed in early autumn | 2.0 |
| Ley plowed in mid-autumn (October–December) | 1.9 |
| Potato | 1.7 |
| Potato followed by catch crop | 1.2 |
| Beet | 0.9 |
| Legumes | 1.3 |
| Flax | 1.3 |

It should be emphasized that basic leaching does not determine the exact quantity of N leached from a given field, because it does not take into account variations of temperature, amount of precipitation and other quantities influencing N leaching from a specific measurement year. Despite these simplifications, basic leaching calculations can help farmers better understand what factors affect N leaching and what actions they can take to reduce it.

## Factor B—type of crop grown in the previous season

The highest N leaching occurs in autumn and winter, that is, at the beginning of each crop year. It is mostly determined by the way in which the field was used in the previous crop year. Thus, crops grown in the previous crop rotation also influence the level of N leaching in the current crop cycle (Table 2).

So if new crops are sown in the autumn, N leaching will decrease, which must be taken into account when estimating the losses. Where temporary grassland is plowed in spring before a new crop is introduced, particular attention should be paid and the relevant coefficient in Table 2 should be multiplied by 1.5. Data from Table 2 cannot be treated only as crop-specific leaching values. For example, N leaching rates in cases such as fodder crops, fallow land, sugar beet and postharvest crops include corrections (adjustments) related to other factors contributing to the reduction of N leaching, for example, late plowing, plow-less tillage.

**Table 3 Factor estimating effect of soil tillage on N basic leaching.** Source: *Aronsson & Ulén (2013)* based on *Aronsson & Torstensson (2004)* and *Hoffmann (1999)*.

| Soil tillage | Factor |
|---|---|
| In early autumn (August–September) | 1.0 |
| Late autumn (October–December) | 0.8 |
| No plowing in the autumn | 0.7 |

## Factor C—soil tillage

Frequent tillage and the associated soil mixing stimulate the release of nitrate N form from the soil, especially if the tillage is carried out at the beginning of autumn. In case of delay or failure to carry out cultivation operations in autumn, nitrate leaching is reduced. Therefore, a coefficient from Table 3 must be used, taking into account the date of plowing in the previous year. If a perennial crop is grown in the field for fodder, the coefficient from the row "No plowing in the autumn" must be used. In the case of potatoes, beet and root crops, it should be assumed that harvesting means the same as soil tillage in late autumn.

## Factor D—application of organic fertilizers

If manure is applied in autumn, some of its N content will be leached. Moreover, with fertilizer, both plant available (mineral) and unavailable (organic) N are introduced into the soil, and the release of mineral N from the latter is not always synchronized with the uptake cycle of the plants. This means that the risk of N leaching increases slightly even after spring application. As shown in Table 3, under the spring application of manure and liquid fertilizers, N leaching is only slightly higher than when only mineral fertilizers in balanced doses are applied. After the application of organic fertilizers in autumn, the leaching is greater than after the application of mineral fertilizers. Slurry (livestock urine with a possible small amount of feces and/or water; contains on average 1–3% of dry matter) consists mainly of plant-available ammonium N, so its fertilizing effect can be compared to that of mineral N fertilizers. Solid manure, on the other hand, contains almost exclusively N in organic form (*Font-Palma, 2019*). Therefore, the release of mineral N from solid manure can be slower than from liquid organic fertilizers (*Antil et al., 2005*). Probably the most favorable way to use manure for N leaching is in spring rather than in autumn. There are discrepancies in the permissible date of application of fertilizers, but the provisions in this respect should be strictly observed (organic fertilizers in liquid and solid form should be applied in the period from 1st March to 30 November, except for fertilizers used in crops under protection, i.e., in greenhouses).

## Factor E—excess N leaching

When the field is fertilized with natural or mineral fertilizers at doses appropriate to the nutritional requirements of the crops grown, N leaching may be considered to be low. If too much fertilizer is applied, the leaching will increase, although an overdose of fertilizer is not intentional. Such a situation is possible during the summer drought when small plants cannot fully benefit from the N introduced with the fertilizers in spring

and early summer. When estimating whether, and if so, too much N was applied on the field, it is appropriate to start by estimating the amount of crop-available N that remained from the previous growing season, that is, the total amount of mineral N supplied by mineral and/or natural fertilizers, and to add the amount of predicted additional N leaching losses due to exceeding the optimum fertilizer application rate for average yields on different soils (expressed in kg N·ha$^{-1}$). In this way, a sum of leaching is obtained. The amount of N applied should be compared with the recommended N dose needed to obtain planned yield of cultivated plants. A good source of information on nutrient requirements of plants is the Program of measures to reduce pollution of waters with nitrates from agricultural sources and to prevent further pollution (*Ministry of Agriculture & Rural Development of Poland, 2018*).

The N load applied is the sum of the amount of N from mineral fertilizer and the expected (approximately) amount of N contained in the natural fertilizers used for cultivation. If the actual amount of N is greater than the recommended amount, refer to Table 5 for the additional N leaching rate.

## Calculations—total N leached from field

The first step in calculating the total N leached from the field (see Fig. 3) is to determine the extra N leaching from Table 5.

It is necessary to calculate the fertilizer intensity first as:

$$I = \frac{\text{TN}}{A}, \quad \text{TN} = \sum_f m_f \cdot c_f,$$

where $I$ is the fertilizer intensity (kg N·ha$^{-1}$), TN is the total N load applied to the field (kg N), $A$ is the area of the field (ha), $m_f$ and $c_f$ are mass of fertilizer (kg) and N content in specific fertilizer respectively, $f$ indexes the fertilizers used in the field. It should be stressed that in the presented method the forms of nitrogen applied with mineral fertilizers are not differentiated and the exact date of application is not taken into account. This means that the method is susceptible to further improvements as the precise determination of the impact of these factors can significantly improve the values of the estimated quantities. In the next step, the excess over the recommended fertilizer intensity should be calculate as:

$$\text{Exc} = I - R \cdot C,$$

where Exc is the excess over the recommended fertilizer intensity (kg N·ha$^{-1}$), $R$ is the recommended N load per tonne of product (kg N·tonne$^{-1}$), $C$ is the expected crop (tonnes ha$^{-1}$). Depending on the value of Exc, for a given soil type, the appropriate value of estimated extra N leaching $E$ is now selected from Table 5. Finally, the total N leaching from the field is calculated as:

$$\text{TNL} = A \cdot B \cdot C \cdot D + E,$$

where TNL is the total N leaching from field (kg N·ha$^{-1}$), $A$ is basic leaching (kg N·ha$^{-1}$) from Table 1, $B$ is the factor affecting basic leaching depending on the crop yield in the

**Table 4 Factor for additional N leaching losses compared with basic leaching depending on manure type.** Based on an application rate of 20–40 tonnes ha$^{-1}$. Source: *Aronsson & Ulén (2013)* based on *Aronsson & Torstensson (2004)* and *Hoffmann (1999)*.

| Type of manure | Autumn | Spring |
|---|---|---|
| Solid manure | 1.15 | 1.10 |
| Slurry | 1.30 | 1.10 |

**Table 5 Estimated extra N leaching (kg N·ha$^{-1}$) for different soil types and the amount by which the recommended fertilizer doses have been exceeded.** Source: *Aronsson & Ulén (2013)* based on *Aronsson & Torstensson (2004)* and *Hoffmann (1999)*.

| Excess over the recommended fertilizer intensity (kg N·ha$^{-1}$) | Sandy soil | Loamy soil | Clay soil | Organic soil |
|---|---|---|---|---|
| 10–20 | 3 | 2 | 2 | 3 |
| 20–30 | 6 | 4 | 4 | 6 |
| 30–40 | 10 | 5 | 5 | 10 |
| 40–50 | 16 | 7 | 7 | 16 |
| 50–60 | 22 | 8 | 8 | 22 |

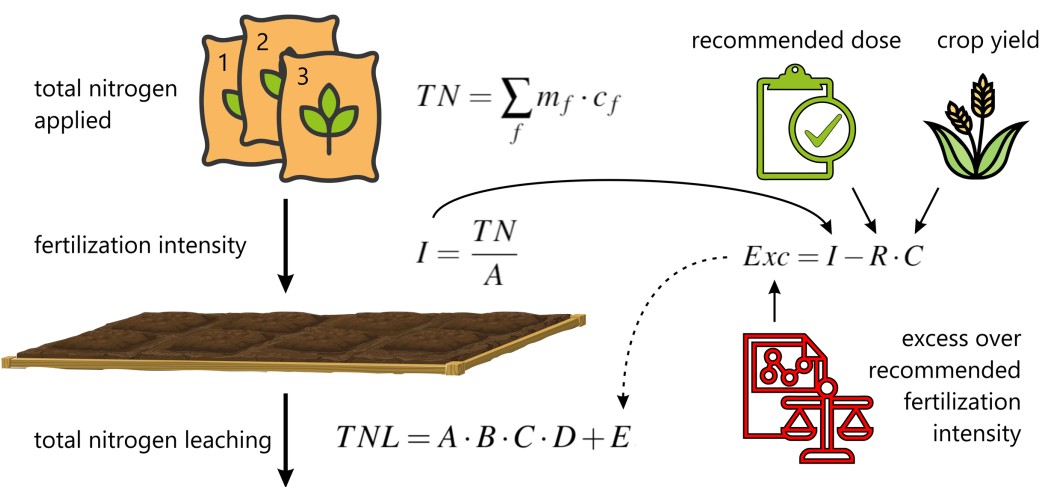

**Figure 3 Scheme of total N leaching from the field calculations.**

previous year from Table 2, $C$ is the factor estimating the effect of soil tillage on N basic leaching from Table 3 and $D$ is the factor for additional N leaching losses compared with basic leaching depending on manure type from Table 4.

## Opinion poll

An opinion poll was conducted on 31 farms within the Puck Commune, which is approximately 3.6% of all farms in this Commune. Field experiments were approved by the

Head of the Puck Commune. Farmers provided the following data for all their fields in the survey:

- soil type (determination of factor A)
- type of crop (determination of factor B)
- date of plowing (determination of factor C)
- information on manure (determination of factor D)
- mass of the product (determination of factor E)
- field area (determination of factor E)
- types and amounts of mineral fertilizers applied on the field (determination of factor E)

## RESULTS

### N leaching calculator

Within the Water PUCK project, a website in the form of an interactive calculator to assist farmers in determining the quantity of N leaching from the field was developed. Access to the calculator is through the main website of the project www.waterpuck.pl through the "Services" tab.

The method of calculating N leaching from an agricultural field described in this paper has been implemented as a website's back-end. After entering the correct input data, the result is refreshed immediately.

The user can easily enter the same information as collected in opinion polls into the leaching calculator (see Fig. 4). Entering data is very intuitive and the result is refreshed on the fly. As a result, the farmer, agricultural adviser or other interested parties can quickly and easily obtain information about:

- basic N leaching (kg N·ha$^{-1}$),
- total mass of N applied (kg N),
- modified N leaching (kg N·ha$^{-1}$),
- crop yield (tonnes ha$^{-1}$),
- fertilization intensity (kg N·ha$^{-1}$),
- extra N leaching (kg N·ha$^{-1}$),
- total N leaching (kg N·ha$^{-1}$),
- total N leached (kg N).

Using the N leaching calculator described here should help farmers to choose the right dosage of N-containing fertilizers to be applied on the field. In addition, the user of the calculator can check what effect the use of natural fertilizers will have on the N leaching. It also informs which fertilization practices increase the risk of excessive leaching of N.

### Surface area of the studied fields

The Puck Commune has the area of 24,266 ha (242.6 km$^2$), which is 1.33% of the area of Pomeranian Voivodeship. Agricultural land is 61% of the Commune's area, including

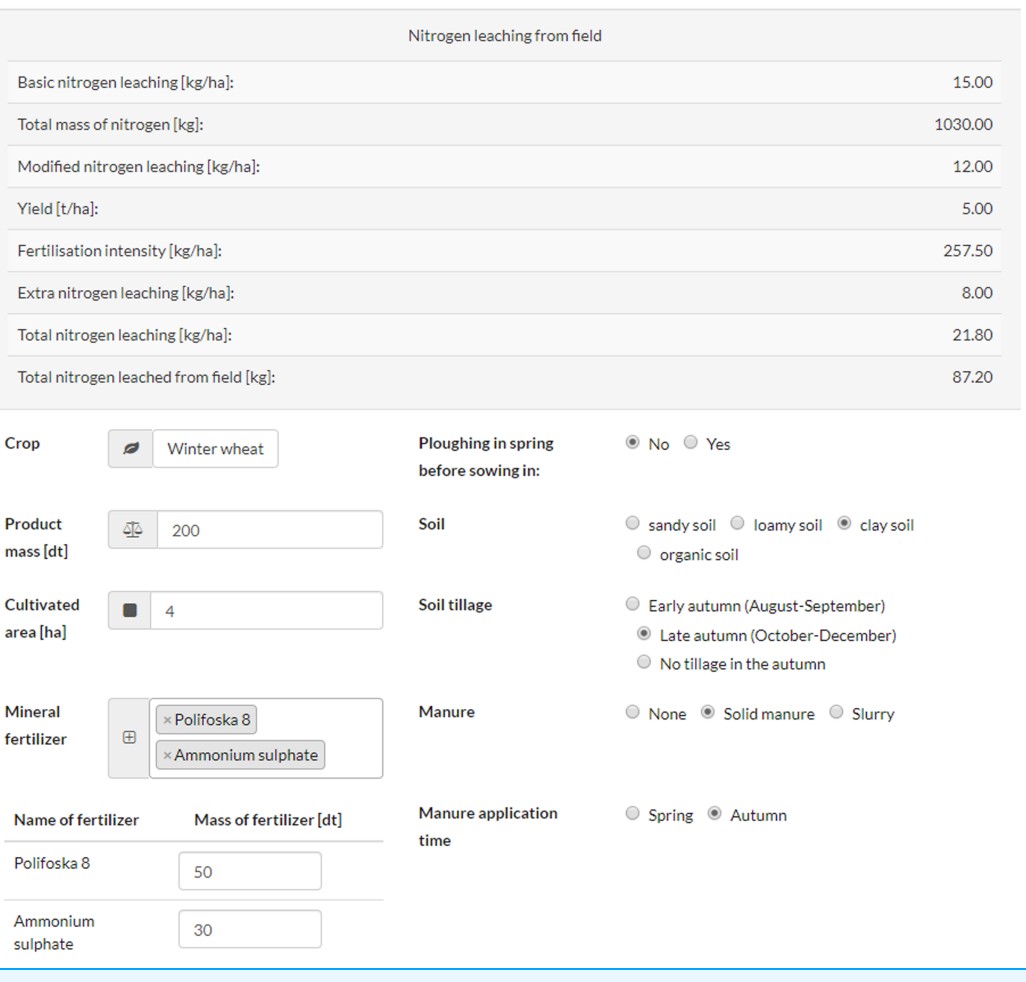

**Figure 4 Calculating load of N leaching from cultivated field (website snapshot).**

72.7% of arable land, 19.2% of meadows, 0.2% of orchards and 4.4% of pastures. Forests are 31.2% of the Puck Commune's area. The area of 291 studied fields varies from 0.1 to 25 ha with a median of 2.3 ha. The distribution of the size of the fields according to the type of crop is shown in a box diagram (see Fig. 5). On the vast majority of agricultural fields ($n = 182$) cereals (wheat, rye, oats, barley, triticale, grain mixtures) are grown and a median area of these fields is equal to 2.25 ha. The second crops with the highest number of fields are fodder crops (silage maize, grass mixtures on arable land) ($n = 55$) with a median area equal to 2.50 ha. Oilseeds (colza) are grown on 30 fields with a median area of 2.32 ha, root crops (potatoes) on 19 with a median area of 0.60 ha, legumes (field bean, lupin, field pea) on 4 with a median area of 4.59 ha and textile crops (linum) are grown on only one field of 5.00 ha.

The total area of all studied fields is equal to 956.74 ha which is about 6.5% of total agricultural land of the Puck Commune. Share of individual crops in the total studied area is presented in Fig. 6. Cereals are grown on more than 60% of the studied area, fodder
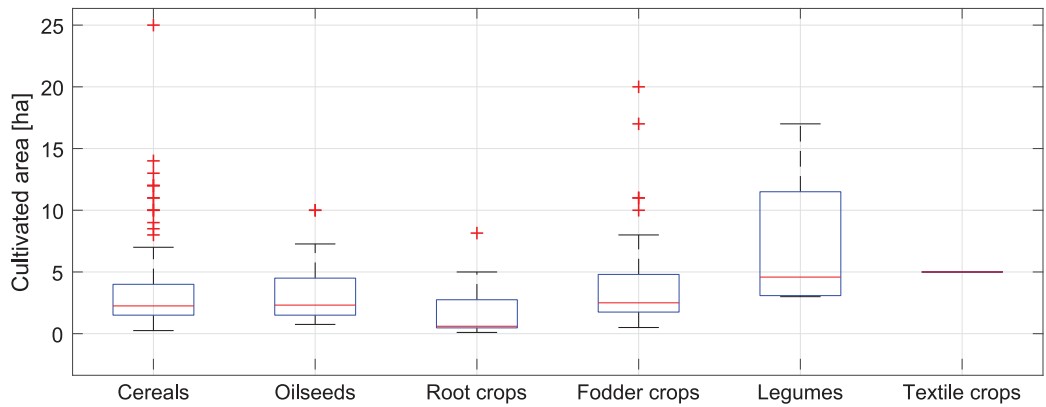

**Figure 5** Box plot of the fields' area of cultivated crops on the studied farms in the Puck Commune in 2018.                                                         

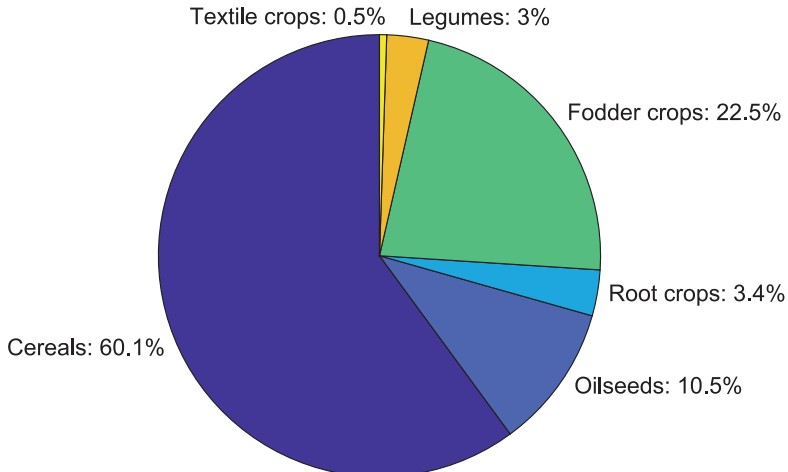

**Figure 6** Share of individual crops area in the total cultivated area in 2018.

crops on 22.5%, oilseeds on 10.5%, root crops on 3.4%, legumes on 3% and textile crops on 0.5%.

## Basic leaching and its modifications

Clay soils with 15 kg N·ha$^{-1}$ of basic leaching (Table 1) are 47.5% ($n = 140$) of the surface area of all studied fields, loamy soils with 20 kg N·ha$^{-1}$ of basic leaching are 45.7% ($n = 134$) and sandy together with organic soils (15 kg N·ha$^{-1}$ of basic leaching) are 6.8% ($n = 17$) of the surface area of all studied fields. Table 2 shows that the type of crop cultivated in the previous year may have the greatest influence on the change in basic leaching and its modifications may range from −40% to 100% of the original value. The number of fields with a specific modification of base leaching is presented in Table 6.

Another factor that may influence the basic leaching is the soil tillage time. According to Table 3, the plowing time can change the basic leaching even up to −30% (if no plowing is done at all). Table 7 shows the number of fields depending on the plowing time.

**Table 6 Number of fields and their total area with a specific modification of base leaching caused by the type of crop from the previous year.**

| Basic leaching modification | −30% | −20% | −10% | 0% | +10% | +20% | +30% | +70% | +100% |
|---|---|---|---|---|---|---|---|---|---|
| Number of fields | 1 | 25 | 26 | 170 | 23 | 10 | 16 | 15 | 5 |
| Total area (ha) | 2.00 | 122.72 | 74.51 | 552.52 | 65.37 | 33.76 | 60.17 | 30.39 | 15.30 |

**Table 7 Number of fields and their total area according to soil tillage time.**

| Soil tillage (basic leaching modification) | Number of fields | Total area (ha) |
|---|---|---|
| Early autumn (0%) | 32 | 120.66 |
| Late autumn (−20%) | 98 | 331.01 |
| No plowing in the autumn (−30%) | 161 | 505.07 |

**Table 8 Number of fields and their total area with specified natural fertilization.**

| Application time and type of manure (basic leaching modification) | Number of fields | Total area (ha) |
|---|---|---|
| No manure application (0%) | 182 | 636.14 |
| Spring—solid manure and slurry (+10%) | 63 | 176.11 |
| Autumn—solid manure (+15%) | 43 | 131.49 |
| Autumn—slurry (+30%) | 3 | 13.00 |

The third and last factor influencing tfhe basic leaching rate is the application of natural fertilizers. In the case of spring natural fertilizer application, basic leaching is modified by +10% regardless of the type of fertilizer. In the case of natural fertilization in autumn, the use of solid manure increases the basic leaching by 15%, while the use of slurry increases the basic leaching by 30%. The categorization of fields by natural fertilization type is shown in Table 8.

It should be emphasized that the change in basic leaching is the product of all three factors analyzed above. Lack of autumn plowing or late autumn plowing can only reduce the amount of basic leaching. However, both the type of crop cultivated in the previous year and the use of manure can potentially increase this value. Thus, the total change in basic leaching due to these factors can range from −51% to even +160% of its initial value resulting from soil type and average annual precipitation in a given region.

## Fertilization intensity

The average value of mineral fertilization intensity calculated as the sum of the total load of N applied to the fields divided by the total area of all fields is equal to 110.94 kg N·ha$^{-1}$. The mineral fertilization intensity for each type of crop is shown in the box plot (see Fig. 7). The highest average intensity of mineral fertilization was applied to oilseeds fields (140.87 kg N·ha$^{-1}$) and the lowest to legumes and textile crops fields (32 and

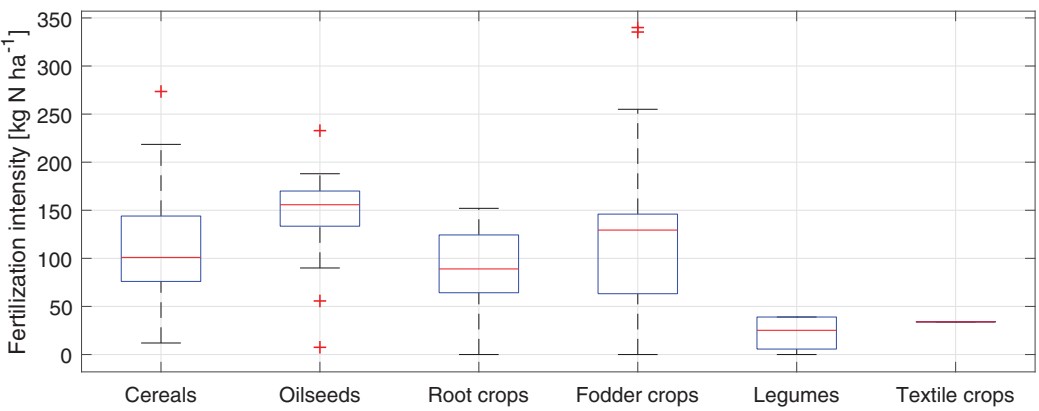

**Figure 7 Box plot of the fertilization intensity of studied fields in the Puck Commune in 2018.**

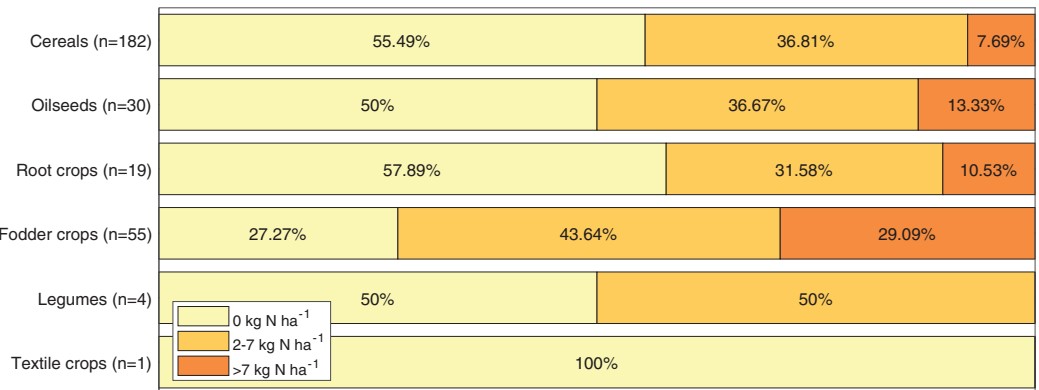

**Figure 8 Extra N leaching from studied fields in the Puck Commune in 2018.**

34 kg N·ha$^{-1}$ respectively). The most intensively fertilized fields (about 340 kg N·ha$^{-1}$) were cultivated with fodder crops.

It should be noted that a large variation in the intensity of fertilization within a given type of crop does not necessarily mean that the intensity of fertilization deviates strongly from the recommended dose, but may result from the different N demand of plants included in a particular crop group.

## Extra N leaching from field

For all 291 studied fields, on the basis of calculations of exceeding the recommended fertilization intensity and data from Table 4, an estimated value of the extra N leaching was determined. For almost half of all fields (49.8%) the extra N leaching is equal to 0 kg N·ha$^{-1}$. For 37.8% of the fields, the extra N leaching value is between 2 and 7 kg N·ha$^{-1}$. In the remaining 12.4% of the fields, the value of the extra N leaching exceeds 7 kg N·ha$^{-1}$. The amount of extra leaching due to the type of plant was presented as a bar chart (see Fig. 8).

**Table 9 Number of fields with specified extra N leaching according to different soil types.**

| Extra N leaching | Loamy soil | Clay soil | Sandy and organic soils |
|---|---|---|---|
| 0 kg N·ha$^{-1}$ | 60 | 74 | 11 |
| 2–7 kg N·ha$^{-1}$ | 57 | 49 | 4 |
| >7 kg N·ha$^{-1}$ | 23 | 11 | 2 |

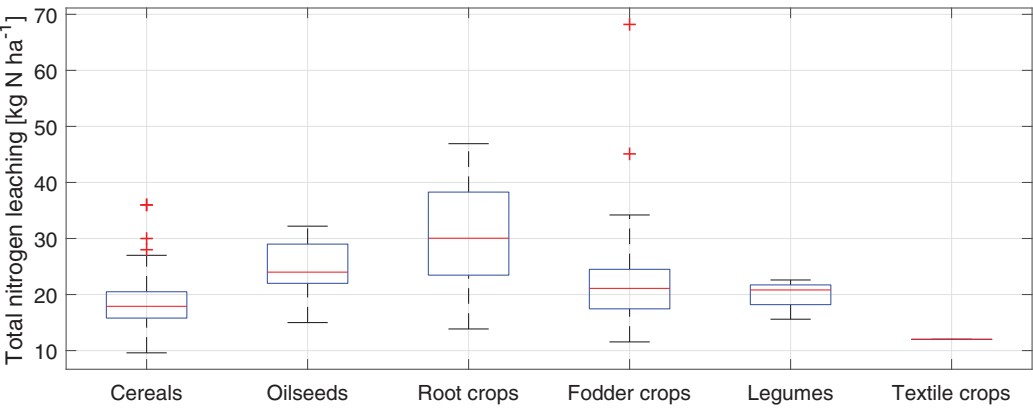

**Figure 9 Box plot of the total N leaching from study fields in the Puck Commune in 2018.**

The extra leaching of N depends on the excess over recommended fertilization intensity and the soil type on which the plant is cultivated. The higher the excess of the actual fertilizer intensity over the recommended fertilizer intensity, the greater the extra N leaching from the field is (Table 5). It is also worth comparing how the extra N leaching from the field varies due to the soil type (i.e., whether farmers apply higher than recommended doses on specific soil types). This comparison is presented in Table 9 and shows that such a relationship does not exist (i.e., the distribution of extra N leaching depending on the soil type is similar to the collective distribution for all fields).

## Total N leaching from field

The total estimated N leaching from studied fields varies from 4.0 to 68.2 kg N·ha$^{-1}$ with a median of 19.8 kg N·ha$^{-1}$. The distribution of the total N leaching from field according to the type of crop is shown in a box diagram (see Fig. 9).

The highest average total leaching of N (weighted by fields' surface areas) is for fields cultivated with root crops (about 33 kg N·ha$^{-1}$) and the lowest for the field cultivated with textile crop (12 kg N·ha$^{-1}$).

## DISCUSSION

In the examined sample of fields, the highest percentage are fields cultivated with cereals (over 60%) while the lowest percentage are fields cultivated with legumes and textile crops (3% and 0.5% respectively). Taking into account all three factors that influence the basic leaching, that is, the type of crop cultivated in the previous year, the time of soil tillage

and the application of natural fertilizers, we can see that the most dominant factor in the examined sample is the time of soil tillage which decreases basic leaching by 30% for more than half of the studied fields. For nearly 60% of the fields, the basic leaching is not changed by the crop type in the previous year, nor is it changed for more than 60% when it comes to natural fertilizer application. Furthermore, a change of basic leaching due to no plowing or late autumn plowing reduces the average basic leaching of N from the fields by approximately 26% which points to very good agricultural practices on soil tillage in the studied region. The amount of basic leaching increases on average by about 12.5% by applying natural fertilizers and, on average, less than 6% by the type of crop cultivated in the previous year.

The average value of mineral fertilization intensity in the studied sample (about 110 kg N·ha$^{-1}$) is higher than Poland's average (80 kg N·ha$^{-1}$) while in other countries of the Baltic Sea region these values are around 30 kg N·ha$^{-1}$ in Sweden and Estonia, over 100 kg N·ha$^{-1}$ in Norway, c.a. 80 kg N·ha$^{-1}$ in Denmark and around 75 kg N·ha$^{-1}$ in Germany (*European Environment Agency, 2018*). A recent study conducted by *Wojciechowska et al. (2019)* aimed at examining loads of N and P released into the Puck Bay from three small first-order agricultural watersheds showed that the mean total N concentrations in the analyzed watercourses were similar to other rivers in central Europe with medium-intensive agricultural land use in the catchments. In the mentioned paper correlation was confirmed between precipitation and concentrations of nutrients in watercourses, pointing out the need for measures counteracting nutrient losses through leaching and erosion.

For almost half of all fields (49.8%) the extra N leaching is equal to 0 kg N·ha$^{-1}$ which means that for the crops grown on these fields the recommended fertilizer doses have not been exceeded. However, there are fields (12.4%) where the extra N leaching exceeds 7 kg N·ha$^{-1}$ and here is a possibility for the agricultural advisers to take action to improve the situation by consulting with the farmers cultivating these fields.

The average (weighted by the surface area of the fields) of the basic leaching of N for the studied sample resulting from the type of soil and precipitation is equal to 18.3 kg N·ha$^{-1}$. While the average basic N leaching modified by factors resulting from the type of crop cultivated in the previous year, the time of soil tillage and the application of natural fertilizers is equal to about 17.5 kg N·ha$^{-1}$ which suggests good agricultural practices due to mentioned factors. However weighted average of total N leaching for the studied field sample is about 20.3 kg N·ha$^{-1}$ (it is greater than the median of the sample, which suggests slightly higher N leaching from relatively larger fields). Therefore, the average total N leaching is about 16% higher than the average modified basic leaching from field and it is caused by exceeding the recommended doses of mineral fertilizers.

Considering the quantity of N leaching from agricultural fields with particular types of crops, it was arranged in the following order: root crops > oilseeds > fodder crops > legumes > cereals > textile crops. Thus, the type of crop, according to what *Simmelsgaard (1998)* stated, is a key factor in shaping nitrate leaching. The largest N losses by leaching were recorded on fields where root crops, especially potatoes were grown. Use of these crops has a high N leaching potential (*Venterea, Hyatt & Rosen, 2011*) which is

related to their relatively shallow root system and high demand for N fertilizers. In literature, there are data showing that N leached from fields where potatoes were grown can reach 143 kg N·ha$^{-1}$ (*Jégo et al., 2008*). At the other end of the spectrum, relatively low quantity of N leaching (apart from N leaching from textile fields which accounts for a small share in the structure of crops) was recorded from fields occupied by cereals. Among them, winter cereals were dominant. To some extent, this state may be explained by the fact that winter cereal species as cover crops have a possibility of capturing N excess and reducing the N leaching by recycling nutrients between autumn and spring seasons (*Brandi-Dohrn et al., 1997*). *Meisinger & Ricigliano (2017)* have shown in this area that winter cereal cover can reduce N leaching by 95% in a dry year and by 50% in a wet year compared to N leaching from uncovered crop fields.

It is difficult to compare estimated N leaching losses in quantitative terms with the results of other studies due to the multitude of natural and anthropogenic factors—often very specific for a given area. As an example, it is worth mentioning that in slightly similar conditions to the Puck Commune, in southwest Sweden N leached (from sandy loam soil) in a mild winter under wheat and oilseed rape amounted to 35–94 kg N·ha$^{-1}$ and 16–23 kg N·ha$^{-1}$, respectively. In cold winter, by contrast, N leaching levels were similar for all crops, at 32–58 kg N·ha$^{-1}$ (*Engström et al., 2011*). These values were higher than the amounts estimated for the tested fields in the Puck Commune. The average N leached from the study area (203 kg N·ha$^{-1}$) was within the lower range of annual losses of nitrates from arable land in southern Sweden at the end of the 20th century which were set at 15–45 kg N·ha$^{-1}$ (*Stenberg et al., 1999*).

## CONCLUSIONS

The interactive N leaching calculator presented at work is a tool that allows farmers to enter data on their agricultural practices in a simple and intuitive way and that displays the results of calculations of the estimated quantity of N leaching in real time. By using a calculator, farmers can also simulate the impact that a change in their current practices will have on N leaching, and thus on soil quality and potentially higher yields in the future. At a time when agriculture is aimed to a massive scale crop cultivation where fertilization and plant protection techniques are extensively used to maximize production efficiency, particular attention should be paid to the risks associated with nutrient leaching. Among these threats, the potential risk of water pollution is particularly important. Further research should be carried out and as simple to implement as possible solutions should be created for farmers, which will ensure a significant reduction in the amount of nutrient leaching from agricultural fields. Forward-looking implementations and perspectives that can improve the quality of surface runoff receivers from fields and prevent erosion include all kinds of Green Infrastructure applications such as constructed wetlands and buffer strips along river beds.

## ACKNOWLEDGEMENTS

The authors of this article would like to thank all the farmers who agreed to take part in the research.

### Funding

This work was supported by the National Centre for Research and Development of Poland within the BIOSTRATEG III program No: BIOSTRATEG3/343927/3/NCBR/2017. The funders had no role in study design, data collection and analysis, decision to publish, or preparation of the manuscript.

### Grant Disclosures

The following grant information was disclosed by the authors:
National Centre for Research and Development of Poland: BIOSTRATEG3/343927/3/NCBR/2017.

### Competing Interests

The authors declare that they have no competing interests.

### Author Contributions

- Dawid Dybowski analyzed the data, prepared figures and/or tables, authored or reviewed drafts of the paper, and approved the final draft.
- Lidia Anita Dzierzbicka-Glowacka conceived and designed the experiments, authored or reviewed drafts of the paper, and approved the final draft.
- Stefan Pietrzak conceived and designed the experiments, authored or reviewed drafts of the paper, and approved the final draft.
- Dominika Juszkowska analyzed the data, authored or reviewed drafts of the paper, and approved the final draft.
- Tadeusz Puszkarczuk performed the experiments, authored or reviewed drafts of the paper, and approved the final draft.

### Field Study Permissions

The following information was supplied relating to field study approvals (i.e., approving body and any reference numbers):
Field experiments were approved by the Head of the Puck Commune.

### Data Availability

The raw data are available in the Supplemental Files.

### Supplemental Information

Supplemental information for this article can be found online at http://dx.doi.org/10.7717/peerj.8899#supplemental-information.

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
