# Peer review of "Estimation of nitrogen leaching load from agricultural fields in the Puck Commune with an interactive calculator"

_PeerJ, doi:10.7717/peerj.8899_

## Round 0.1 · original submission · Major Revisions

The manuscript addresses an interesting subject and has notable practical implications.

In general, I agree with the two reviewers and recommend taking into account their suggestions.

I am not quite sure that the last two paragraphs of the Introduction should be moved completely to Materials and Methods. The research area must be indicated in Materials and Methods. But the Introduction can include some information about the problem and the current situation in the study area.

The section Materials and Methods contains a substantial part of the study. It describes the method applied to farms in the Puck Commune. A number of Tables present figures used to estimate nitrate leaching from different fields. It seems that this method has not been previously published. If so, the method must be explained in more detail and the figures in the Tables justified. This explanation of the method should constitute a specific section of the manuscript. If the method has already been published, a reference must be given.

See more comments on the annotated manuscript.

Reviewer 1 ·

Basic reporting

See the attachment.

Experimental design

See the attachment.

Validity of the findings

See the attachment.

Additional comments

See the attachment.

Annotated reviews are not available for download in order to protect the identity of reviewers who chose to remain anonymous.

Reviewer 2 ·

Basic reporting

This manuscript is essentially a continuation of the article published in the Peer journal in 2019 (Dzierzbizka et. al., 2019). In the example of the Puck Commune, it is shown how it is possible to calculate nitrogen losses from the agricultural lands located in this area. This type of an article can be attributed to “Case studies” articles, which do not fully meet the requirements of the Peer journal. However, the principles of calculating the nitrogen losses because of leaching that are described in the manuscript are interesting for scientists in other countries, which is why I would like to compare the methods of calculation and improve them.

The manuscript describes the stages of calculating nitrogen loss in some detail and in a sequence, but it does not explain how the correction factors for the main parameters that are affecting the amount of nitrogen leaching were calculated. Were either of these factors calculated on the basis of data collected from the lysimetric and field experiments, or were they obtained by empirical means?
The climatic conditions of the region are not sufficiently described. This makes it difficult to compare the data of this work to the results of studies carried out in other Baltic countries.
The manuscript quotes a few references, especially in the “Discussion” section. The articles cited are mainly presented in the “Introduction” section. Only 2 sources are quoted in the “Discussion” section, therefore this section of the manuscript does not fully meet the requirements of the journal. The article presents 10 figures. Figure 4 in this manuscript is very similar to a figure which has already appeared in a published article in Peer journal (Dzierzbizka et. al., 2019). It also does not fully comply with the requirements of the press for the design of the manuscripts. I believe that the Discussion section should contain more comparisons to the results of other researchers, especially from the Baltic Sea region.

Check the list of references cited in the manuscript. Why is the article by Álvarez et. al (2017) at the end of the list of references?

Experimental design

Line 71-72. The article does not state the exact purpose of the work. Since an adapted mathematical model was used to calculate nitrogen losses, the authors based on this model had only calculated (did not determine) the possible nitrogen losses in the Puck Commune region.

Line 81-82. In my opinion, the text does not describe the beginning of the atmospheric precipitation infiltration accurately. Due to climate change, there may be deviations in the beginning and the ending of infiltration during the time period from October to April. In order to better understand the process of the infiltration of atmospheric precipitation in the region, it is advisable to provide data of the standard climate norm of precipitation and air temperature, as well as to indicate the number of days in the year with negative air temperature, as this affects the infiltration regime.

Line 95. A very old source of reference (1982) was used to describe the effect of legumes on the nitrogen content of the soil. I believe that currently there are many more informative articles supporting this fact.

The calculation of nitrogen losses from leaching is based on the concept of parameters of the basic leaching. However, the manuscript does not describe how the basic values were defined clearly. It is necessary to explain this in the manuscript or there should be a reference to a list of references. Therefore, there are some questions as to why such factors are applied to certain factors.
For example:
a) why is the correction factor to determine the effect of plants on nitrogen leaching for legumes and flax the same (1.3) (Table 2)?
b) Why are there such small differences in factors regarding manure and slurry application in autumn and winter? The main nitrogen leaching occurs in winter, as shown in Figure 3.

In Tables 6-7 it is advisable to indicate not only the number of fields with similar soil and agro technical measurement parameters in the Puck commune, but also the complete area of these fields for certain indicators, as their area can vary considerably. Nitrogen leaching depends on the specific field area, not on the quantity.

Validity of the findings

Lines 291-302 repeat the statistical data on the area of the fields under the different crops and how the usage of fertilizers and the ploughing time of the fields both affect the leaching of nitrogen. There is no discussion and no comparison to the results of other studies. The discussion focuses only on the description of the outcomes.

Conclusion
The conclusions do correspond to the objectives of this work.

Additional comments

The manuscript may be recommended to be published in a Peer journal, but there is a need to expand the discussion on nitrogen leaching in farmland. This manuscript presents only the results of the calculation of nitrogen losses calculated on the basis of the mathematical model in Puck Commune and therefore, this is of interest only to a local region. I realize that this model is designed for the soil and climatic conditions of Poland, but there is a lot of published data that can be comparable to the data that was obtained based on the mathematical model. A more detailed discussion will enable the comparison of processes of the nitrogen leaching and improve the methods of calculation of its losses in the countries of the Baltic region.

---

## Round 0.2 · Minor Revisions

The authors have made the corrections proposed and responded to most comments by both reviewers and mine. I have made some minor comments in the annotated manuscript.

Reviewer 1 ·

Basic reporting

No comments

Experimental design

No comments

Validity of the findings

No comments

Additional comments

The authors of the publication made all the corrections and responded to all my comments. I am satisfied with the answers given. I have no substantive remarks to work, but I am asking the authors for some minor editorial corrections.
Line 126: -1 should be in upper index.
Line 240-244; 288-292; 297-301; 311-315 and Discussion:
I think the authors should use SI unit, i.e. kg N·ha-1, not kg N ha-1.

After correction, I recommend publishing this article in that form.

---

## Round 0.3 · accepted · Accept

It is a pleasure to accept your revised manuscript for publication.